# Shape of a sound wave in a weakly-perturbed Bose gas

O. V. Marchukov[1, 2*], A. G. Volosniev[3],

**1** Institute for Applied Physics, Technische Universität Darmstadt, 64289 Darmstadt, Germany
**2** Department of Physical Electronics, School of Electrical Engineering, Faculty of Engineering, and Center for Light-Matter Interaction, Tel Aviv University, 6997801 Tel Aviv, Israel
**3** Institute of Science and Technology Austria, Am Campus 1, 3400 Klosterneuburg, Austria
* oleksandr.marchukov@tu-darmstadt.de

August 11, 2020

## Abstract

We employ the Gross-Pitaevskii equation to study acoustic emission generated in a uniform Bose gas by a static impurity. The impurity excites a sound-wave packet, which propagates through the gas. We calculate the shape of this wave packet in the limit of long wave lengths, and argue that it is possible to extract properties of the impurity by observing this shape. We illustrate here this possibility for a Bose gas with a trapped impurity atom – an example of a relevant experimental setup. Presented results are general for all one-dimensional systems described by the nonlinear Schrödinger equation and can also be used in nonatomic systems, e.g., to analyze light propagation in nonlinear optical media. Finally, we calculate the shape of the sound-wave packet for a three-dimensional Bose gas assuming a spherically symmetric perturbation.

# 1  Introduction

Time evolution of weakly perturbed quantum gases and liquids is often visualized as the dynamics of collective excitations, e.g., phonons. For example, the response of super-fluid helium-4 to various weak perturbations is interpreted as generation of elementary excitations in the Landau's theory of superfluidity [1–3]. Similar approaches are used to understand cold atoms [4], polaritons [5, 6], and other quantum many-body systems. The general idea is that low-energy perturbations lead to certain occupancies of collective modes, whose dynamics determines the later state of the system. Looking at the problem from another angle, the population of collective modes after excitation carries information about perturbation. This information could potentially be used to study the source of perturbation, as done in acoustic emission testing in classical solids [7]. To exploit this possibility, one first one needs to study theoretically the question *"Can one hear the shape of a drumstick?"*[1], i.e., one needs to understand what information is carried in the sound-wave packet generated by perturbation.

In this work, we use a weakly-perturbed Bose gas to address the question, and investigate the possibility of reconstructing perturbing potential from sound waves. We choose to model the problem using the Gross-Pitaevskii equation (GPE) – the standard tool for studying degenerate Bose gases [9]. Our work focuses on the linear regime of the GPE, which has sound waves as elementary excitations. Nonlinear phenomena supported by the GPE (e.g., solitons, shock waves [10]) are not important for our study, and will be a subject of our future work. For simplicity, we focus on a quasi-one-dimensional Bose gas that can be modelled by a one-dimensional GPE [11–13], and only briefly discuss what happens in higher spatial dimensions.

Our work is summarized in Fig. 1. A static impurity inserted in a homogeneous Bose gas creates a defect in the Bose gas and two sound waves, which contain information about the spatial profile of the impurity. One can learn later properties of the impurity by analyzing the emitted sound. This could allow one to extract properties of the impurity even if its exact location is not known. We illustrate this idea by studying time evolution of the system upon an introduction of a single weakly-interacting impurity of a general kind. The problem is motivated, in particular, by Bose gases with a localized defect [4] or with a massive moving impurity [14].

Our findings are applicable to all systems that are described by the nonlinear Schrödinger equation (NLSE), e.g., to optical pulses propagating inside lossless optical fiber [15], because the Gross-Pitaevskii equation is mathematically equivalent to the NLSE. Furthermore, our results for weak couplings can be applied to other one-dimensional Hamiltonians with similar linear excitations, e.g., to the Fröhlich model with a static impurity (cf. Ref. [16]).

The organization of this paper is as follows. In section 2, we introduce the model (the GPE), which we analyze in the linear regime. In section 3, we compare the results obtained in the linear regime to the numerically exact solution of the Gross-Pitaevskii equation. In section 4, we illustrate our findings for a relevant cold-atom set-up, where the role of perturbation is played by an impurity atom. In section 5, we briefly discuss a weakly-pertubed Bose gas in three spatial dimensions. We conclude in section 6 with a summary of our findings and future directions.

---

[1]We formulate this question in reminiscence of a classic *"Can one hear the shape of a drum?"* (Ref. [8]).

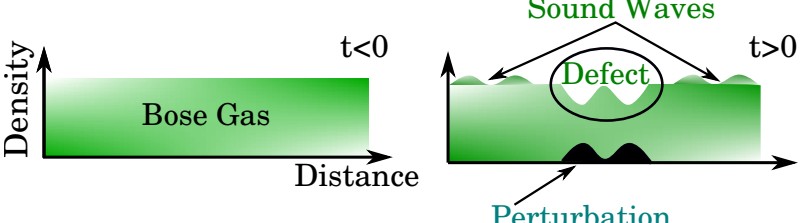

Figure 1: An illustration of the system. A Bose gas which is homogeneous at $t < 0$ is perturbed at $t = 0$. At $t > 0$, the impurity (perturbation) creates a defect in the density of the Bose gas, which resembles the shape of the impurity. Moreover, the impurity generates sound waves, which carry away information about the spatial profile of the impurity.

## 2  Formalism

Our system consists of $N$ repulsively interacting bosons that can be described via the one-dimensional GPE. The system is confined to a ring of length $L$, otherwise it does not experience any external potential at $t < 0$. We focus on the thermodynamic limit $N(L) \to \infty$, assuming homogeneous density $\rho = N/L$ at $t < 0$. At $t = 0$ the system is weakly perturbed, and we study the time evolution at $t > 0$ using the equation

$$i\hbar \frac{\partial \phi}{\partial t} = -\frac{\hbar^2}{2m}\frac{\partial^2 \phi}{\partial x^2} + gN|\phi|^2\phi + \eta V(x)\phi, \tag{1}$$

where $\phi(x, t)$ is the order parameter, $m$ is the mass of a boson, $g$ determines the strength of the interaction between the atoms in the gas, $V$ and $\eta$ define the geometry and strength of the perturbation potential, respectively. For simplicity, we focus on parity-symmetric potentials, i.e., $V(-x) = V(x)$, that are real and decay exponentially fast at infinity. Otherwise, there are no assumptions on the form of $V$, moreover, the generalization for nonsymmetric potentials is straightforward. Note that an important Gaussian perturbation has been extensively studied in Refs. [17–21] – these works provide reference points for our study. The function $\phi$ obeys the initial condition $\phi(x, 0) = 1/\sqrt{L}$. It is periodic, $\phi(x, t) = \phi(x + L, t)$, and it is normalized as $\int |\phi(x, t)|^2 \mathrm{d}x = 1$. For later convenience, we associate a length scale, $l$, with the potential $V$, and define the healing length of the gas as $\xi = \frac{\hbar}{\sqrt{2mg\rho}}$.

We assume that the strength of the perturbation is a small parameter, $\eta \to 0$, that allows us to expand $\phi$ as

$$\phi(x, t) = \frac{1 + \eta f(x, t)}{\sqrt{L}} e^{-i\frac{g\rho}{\hbar}t} + \eta^2 F(x, t) + \dots. \tag{2}$$

We are interested in the evolution of the function $f(x, t)$. To compute it, we study the equation

$$i\hbar \frac{\partial f}{\partial t} = -\frac{\hbar^2}{2m}\frac{\partial^2 f}{\partial x^2} + \rho g(f + f^*) + V(x). \tag{3}$$

Note that the form of the ansatz (2) ensures the non-standard form of the linearized GPE in Eq. (3).

Consider first $V(x) = 0$. In this case, Eq. (3) is well-known: it describes excitations of a uniform Bose gas, and is solved using the Bogoliubov transformation

$$f^{(0)}(x, t) = \int_{-\infty}^{\infty} \mathrm{d}k \left( u_k e^{ikx - i\omega t} - v_k^* e^{-ikx + i\omega t} \right), \tag{4}$$

where $u_k$ and $v_k$ obey the Bogoliubov-de Gennes system of equations:

$$\hbar\omega u_k - \frac{\hbar^2 k^2}{2m} u_k - \rho g(u_k - v_k) = 0,$$

$$\hbar\omega v_k^* + \frac{\hbar^2 k^2}{2m} v_k^* - \rho g(u_k^* - v_k^*) = 0, \tag{5}$$

whose eigenvalues are

$$\omega_k = |k| \left( \frac{\hbar^2 k^2}{4m^2} + \frac{\rho g}{m} \right)^{1/2}. \tag{6}$$

The frequency $\omega_k$ defines the Bogoliubov's excitations spectrum whose relevance for excitations of 1D Bose gases is confirmed by the Bethe ansatz results [22, 23]. Note that for long wave lengths ($k \to 0$) the spectrum is phonon-like

$$\omega_k = c|k|, \tag{7}$$

where $c = \sqrt{\frac{\rho g}{m}}$ is the speed of sound in the gas. There also exist the so-called zero and "lost" modes that are consequences of the Nambu-Goldstone theorem [24, 25] due to the $U(1)$ symmetry breaking [26–30], but their contribution to $f^{(0)}$ is negligible, and we do not consider them here. Equations (5) and (6) allow us to calculate the function $f^{(0)}$:

$$f^{(0)}(x,t) = \frac{1}{2\pi} \int \mathrm{d}k f_k(x,t) e^{ikx}; \qquad f_k = u_k \left( e^{-i\omega_k t} - \frac{\rho g e^{i\omega_k t}}{\hbar\omega_k + \epsilon_k + \rho g} \right), \tag{8}$$

where $\epsilon_k = \frac{\hbar^2 k^2}{2m}$. For later convenience, we have assumed that $u_k$ is real, thus $u_{-k}^* = u_k$. The coefficients $u_k$ are determined by the initial conditions, $f^{(0)}(x, t = 0)$.

We now proceed to the $V(x) \neq 0$ case, for which the solution $f(x,t)$ is written as

$$f(x,t) = f^{(0)}(x,t) + f_{sp}(x), \tag{9}$$

where $f_{sp}(x)$ is a special solution to Eq. (3), which does not depend on time because $V$ does not depend on time. The initial conditions demand that $f^{(0)}(x, t = 0) = -f_{sp}(x)$, which fully determines the function $f^{(0)}$. In order to find $f_{sp}$ we consider the inhomogeneous ordinary differential equation

$$\mathcal{L} f_{sp} = V(x), \tag{10}$$

where $\mathcal{L} = \frac{\hbar^2}{2m} \frac{d^2}{dx^2} - \rho g(I + K)$ is the linear differential operator in Eq. (3), with $I$ and $K$ being the unity and the complex conjugation operators, respectively. We write the Green's function, $G(x, x') = G(x - x')$, of this operator as

$$G(x - x') = -\frac{1}{2\pi} \int \frac{e^{ik(x-x')}}{\epsilon_k + 2\rho g} \mathrm{d}k, \qquad \left[ \mathcal{L} G(x - x') = \delta(x - x') \right], \tag{11}$$

which allows us to solve Eq. (10) as

$$f_{sp}(x) = \int G(x - x') V(x') \mathrm{d}x' \qquad \text{or} \qquad f_{sp}(x) = -\frac{1}{2\pi} \int \mathrm{d}k \frac{\tilde{V}(k) e^{-ikx}}{\epsilon_k + 2\rho g}, \tag{12}$$

where $\tilde{V}(k) = \int V(x) e^{ikx} \mathrm{d}x$ is the Fourier transform of the potential $V(x)$. The inverse Fourier transform is then $V(x) = \frac{1}{2\pi} \int \tilde{V}(k) e^{-ikx} \mathrm{d}k$.

Therefore, Eq. (9) takes the form

$$f(x,t) = \int \mathrm{d}k e^{-ikx} \left[ u_k \left( e^{-i\omega_k t} - \frac{\rho g}{\epsilon_k + \hbar\omega_k + \rho g} e^{+i\omega_k t} \right) - \frac{\tilde{V}(k)}{2\pi(\epsilon_k + 2\rho g)} \right]. \tag{13}$$

Taking into account the initial condition $f(t=0)=0$, we find

$$f_k(t) = \frac{\left[(\epsilon_k + \hbar\omega_k + g\rho)e^{-i\omega_k t} - g\rho e^{i\omega_k t}\right]\tilde{V}(k)}{2\pi(\epsilon_k + \hbar\omega_k)(\epsilon_k + 2g\rho)}. \tag{14}$$

The function $f_{sp}(x)$ is real, since $\tilde{V}(k) = \tilde{V}(-k)$, and $\epsilon_k = \epsilon_{-k}$. Note that in the special case $k=0$ the analyticity of the $f_k$ allows us to evaluate the limit that yields

$$f_0(t) = \frac{\tilde{V}(0)}{4\pi\rho g} - \frac{it}{2}\frac{\tilde{V}(0)}{2\pi\hbar}. \tag{15}$$

This mode is related to the Nambu-Goldstone mode that appeares from the breaking of the translational invariance [31]. The real and imaginary parts of the function $f(x,t)$ are written as

$$\mathrm{Re}(f) = \frac{1}{2\pi}\int dk \frac{\tilde{V}(k)e^{-ikx}}{\epsilon_k + 2\rho g}\left(\cos\omega_k t - 1\right), \tag{16a}$$

$$\mathrm{Im}(f) = -\frac{1}{2\pi}\int dk \frac{\tilde{V}(k)e^{-ikx}}{\epsilon_k + 2\rho g}\left(1 + \frac{2\rho g}{\epsilon_k + \hbar\omega_k}\right)\sin\omega_k t. \tag{16b}$$

We have shown that the knowledge of $f_k$ grants access to $\tilde{V}(k)$, hence, if there is an apparatus to measure the occupation of the excitation spectrum one can learn properties of the perturbing impurity. Next, we analyze $f_k$ for perturbations with long wavelengths, i.e., we focus on the limit $\xi \ll l$ where the GPE works best. In the energy domain, this limit reads

$$\frac{\hbar^2 k_{pert}^2}{2m} \ll g\rho, \tag{17}$$

where $k_{pert} = 1/l$ determines the range of $\tilde{V}(k)$. Equation (17) allows us to simplify $f_k$ and to write the real and imaginary parts of $f$ as

$$\mathrm{Re}(f) \simeq \frac{1}{4\pi g\rho}\int dk\tilde{V}(k)\left(\cos(c|k|t) - 1\right)e^{-ikx}, \tag{18}$$

$$\mathrm{Im}(f) \simeq -\frac{1}{2\pi\hbar c}\int dk\tilde{V}(k)\frac{\sin(c|k|t)}{|k|}e^{-ikx}. \tag{19}$$

It is worthwhile noting that the derivative of $\mathrm{Im}(f)$ is related to the time-dependent part $\mathrm{Re}(f)$. Using the convolution theorem for inverse Fourier transform, we obtain

$$\mathrm{Re}[f(x,t)] \simeq \frac{1}{4g\rho}\left(V(x-ct) + V(x+ct) - 2V(x)\right), \tag{20}$$

$$\mathrm{Im}[f(x,t)] \simeq -\frac{1}{2\hbar}\int_0^t dt'\left(V(x-ct') + V(x+ct')\right). \tag{21}$$

These equations show that two counterpropagating sound waves are formed upon excitation of the Bose gas[2]. For $t \gg ml^2/\hbar$, the zero mode supports a phase difference between parts of the Bose gas, e.g, $\mathrm{Im}(f(|x|\to\infty,t) - f(0,t)) \simeq \tilde{V}(0)/(2c)$.

One can extract information about the perturbing potential by observing the density of the Bose gas. Indeed, the density of the gas $n(x,t) = N|\phi|^2$ is written as $n(x,t) = \rho(1 + 2\eta\mathrm{Re}(f))$ or

$$n(x,t) = \rho + \frac{\eta}{g}\left[\frac{1}{2}V(x-ct) + \frac{1}{2}V(x+ct) - V(x)\right]. \tag{22}$$

---

[2]Note that the imaginary part can also be written as $\mathrm{Im}(f) \simeq \int_{-\infty}^{\infty}\frac{rdr}{4c|r|\hbar}\left[V(x-r-ct) - V(x-r+ct)\right]$.

Note that $\int \mathrm{Re} f \mathrm{d}x = 0$ to ensure correct normalization of $|\phi(x,t)|^2$. In the linear regime, the density of the gas at $t > 0$ is fully defined by $\rho$ and the shape of the perturbation potential. The perturbation creates a stationary defect in the density of the gas given by $V(x)$. It also leads to two waves propagating in the opposite directions with the speed of sound, $V(x \pm ct)$. One can learn about the shape of the impurity by observing the propagation of these sound waves. Furthermore, one can speculate that the running waves can potentially be useful for short-distance communication between different points of the Bose gas provided that $V$ is tailored to the needs of information transfer.

To understand the physics behind the density presented in Eq. (22), a few insights are needed. First of all, the propagation of a sound wave in the one-dimensional Bose gas can be described by the massless (1+1) Klein-Gordon equation[3] whose general (d'Alembert's solution) is $v(x - ct) + w(x + ct)$, where the functions $v$ and $w$ must be determined from the initial conditions. Note that we should expect $v = w$ due to the mirror symmetry of our problem. To find the form of $v$, note that the stationary GPE must describe well the system at around $x = 0$ for $t \to \infty$, i.e., long after the perturbation is tuned on. In other words, the Thomas-Fermi approximation must be valid at $t \to \infty$ close to the impurity potential, which explains the last term in Eq. (22). The three terms ($v(x - ct)$, $v(x + ct)$, and $V$) must all enter in the final result for the density since we work in the linear regime. The function $v$ can then be found from the initial condition $2v(x) - V(x) = 0$, giving a clear explanation for the form of $n(x,t)$.

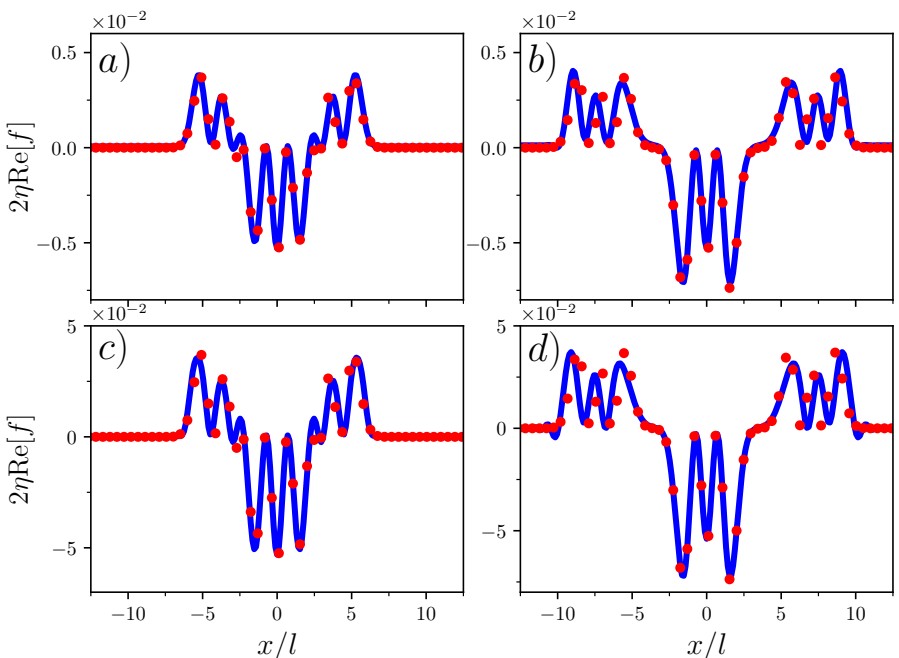

Figure 2: Time evolution of the density of the Bose gas, $n(x,t)/\rho - 1$. A solution of the GPE is shown by the solid blue curves; the linearized solution is presented using red dots. The perturbation potential is given by Eq. (23). Numerical calculations are implemented for $N = 1500$, $g = 1.0 \left(\frac{\hbar}{ml}\right)$, and different values of $\eta$. The density of the Bose gas at $t = 0$ is $\rho = 50/l$, which ensures that $\xi \ll l$. Panels (a) and (b) show the snapshots at $t \approx \{3.5 \left(\frac{ml^2}{\hbar}\right), 7.1 \left(\frac{ml^2}{\hbar}\right)\}$ for $\eta = 0.5$. Panels (c) and (d) demonstrate the densities at $t \approx \{3.5 \left(\frac{ml^2}{\hbar}\right), 7.1 \left(\frac{ml^2}{\hbar}\right)\}$ for $\eta = 5$.

---

[3]The Klein-Gordon equation follows from Eq. (3).

# 3   Exact solution versus linearization

The linearized solution obtained above neglects the higher order terms in the expansion of Eq. (2). To estimate the accuracy of this approximation, we compare the linearized solution to a numerical solution of the GPE. For the sake of discussion, we consider the perturbation potential,

$$V(x) = \frac{\hbar^2}{ml^2\sqrt{\pi}} \left( 2\left(\frac{x}{l}\right)^2 - 1 \right) \exp\left[ -\frac{1}{2}\left(\frac{x}{l}\right)^2 \right], \tag{23}$$

inspired by the second excited eigenstate of a harmonic oscillator. This choice is made to ensure that the numerical solution of the GPE would be well-behaved and would clearly show the shape of the running waves.

In Fig. 2, we compare the normalized density of the gas, $n(x,t)/\rho - 1$, to $2\eta\mathrm{Re}(f)$ from Eq. (22). The numerically 'exact' density is given by $L|\phi^{(num)}|^2 - 1$ where $\phi^{(num)}$ is a solution of the GPE. This solution is obtained using the pseudospectral method (equipped with the fast Fourier transform routine) for space discretization, in combination with the Runge-Kutta time-stepping scheme [32]. In Fig. 3, we compare the phase of the numericall 'exact' solution to that obtained by linearization.

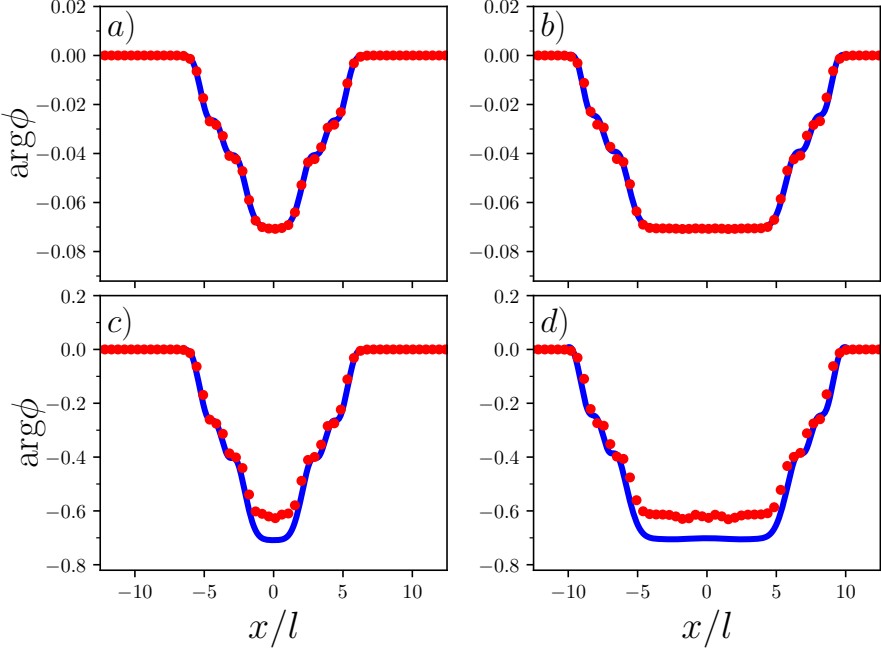

Figure 3: Time evolution of the phase of the Bose gas, $\arg(\phi)$. A solution of the GPE is shown by the solid blue curves; the linearized solution is presented using red dots. The perturbation potential is given by Eq. (23). Numerical calculations are implemented for $N = 1500$, $g = 1.0 \left(\frac{\hbar}{ml}\right)$, and different values of $\eta$. The phase is taken relative to the phase at $x = -L/2$ to ensure that it is well-defined. Panels (a) and (b) show the snapshots at $t \approx \{3.5\left(\frac{ml^2}{\hbar}\right), 7.1\left(\frac{ml^2}{\hbar}\right)\}$ for $\eta = 0.5$. Panels (c) and (d) demonstrate the phases at $t \approx \{3.5\left(\frac{ml^2}{\hbar}\right), 7.1\left(\frac{ml^2}{\hbar}\right)\}$ for $\eta = 5$.

Figures 2 and 3 show that the results obtained via linearization agree well with our numerical solution of the GPE. Linearization captures the behavior of the solution even

after a long evolution time, when the running waves are clearly separated. For strong interactions, the nonlinear effects of the GPE muddle the quantitative comparison (especially for the phase of the gas); still a qualitative agreement is observed. Note that the shape of the sound waves in Fig. 2 resembles the shape of the impurity even for rather large values of the perturbation parameter, $\eta$, for which the density variation can be approximately 7%. The effect of this size is within reach of current experimental setups, allowing one to extract properties of the impurity from the generated sounds waves. The presence of the left- and right-moving wave packets is useful for averaging out random noise. Note also that one can study the impurity from the static defect created by the impurity in the density of the Bose gas, provided that the position of the impurity is known.

# 4 A mobile impurity in a Bose gas

Finally, we discuss a system where observation of the sound waves examined above may provide a valuable tool for a weakly destructive measurement of the perturbation's properties. Such systems can be Bose gases with localized or slowly moving defects [4, 14], or systems with mobile impurities.

We choose to consider a homogeneous Bose gas with a mobile impurity atom trapped by an external potential. A mobile impurity could be used to study properties of the Bose gas [33–35], to simulate Bose polarons [36], and to store and process quantum information [37, 38]. These applications have already motivated a number of works to study the quench dynamics of impurities in similar setups [36, 39–43]. In this work, we investigate the corresponding time dynamics of the Bose gas.

We assume that the impurity occupies a certain state of the harmonic trap, and argue that information about this state can be extracted in a weakly destructive manner by observing the density of the Bose gas following a sudden change of the boson-impurity interaction. For simplicity, we focus on an impurity that is either in the ground, $|g\rangle$, or in the first excited, $|e\rangle$, states. A more challenging measurement of a general quantum state: $|g\rangle + \mathcal{A}|e\rangle$, where $\mathcal{A}$ is some complex number, requires knowledge of both the density and the phase of the Bose gas. A corresponding discussion is left for future studies.

To study the dynamics of the impurity we employ a strong coupling approach [44–46], which is based on the Hartree approximation to the wave function. The system (Bose gas plus impurity) is described within this approach by the system of coupled equations

$$i\hbar\frac{\partial\phi}{\partial t} = -\frac{\hbar^2}{2m}\frac{\partial^2\phi}{\partial x^2} + gN|\phi|^2\phi + g_{ib}|\psi|^2\phi, \tag{24}$$

$$i\hbar\frac{\partial\psi}{\partial t} = -\frac{\hbar^2}{2m}\frac{\partial^2\psi}{\partial x^2} + \frac{\hbar^2x^2}{2ml^4}\psi + g_{ib}N|\phi|^2\psi, \tag{25}$$

where $\phi$ is the order parameter of the Bose gas, and $\psi$ describes the impurity; $g_{ib}$ is the strength of the boson-impurity interaction. The impurity is trapped by an external harmonic oscillator, $\frac{\hbar^2x^2}{2ml^4}$. For simplicity, we assume that the impurity and a boson are of equal masses. We again consider $\xi \ll l$, which means that the density distortion created by the impurity has a range, which is smaller than the oscillator length. Interestingly, this condition is also required to define a one-dimensional Bose polaron [43, 47, 48].

In the limit of small values of $g_{ib}$, Eq. (24) maps onto Eq. (1) with

$$\eta V(x) = g_{ib}e^{-x^2/l^2}/(\sqrt{\pi}l) \tag{26}$$

for the ground state and

$$\eta V(x) = 4g_{ib}x^2e^{-x^2/l^2}/(\sqrt{\pi}l^3) \tag{27}$$

for the excited state. The Bose gas is homogeneous at $t < 0$, which means that in the leading order in $g_{ib}$, only the phase of the impurity atom is affected. By changing $g_{ib}$ one creates a sound wave in the Bose gas with amplitude proportional to $g_{ib}$, whereas the corresponding change of the state of the impurity is of the order $g_{ib}^2$. Let us show that the sound wave indeed contains information about the state of the impurity. To this end, we solve Eqs. (24) and (25) numerically and via linearization. The comparison is presented in Fig. 4. We see that numerical results agree well with linearization. As expected, the shape of the sound wave is given by the state of the impurity. If the impurity is used to store information in the form of either $|g\rangle$ or $|e\rangle$, then the read-out of this information can be achieved from the sound waves emitted upon the change of $g_{ib}$. The measurement process is weakly destructive, since the state of the impurity is perturbed as $\sim g_{ib}^2$ upon the change of $g_{ib}$, and the observation of the density of the Bose gas may (in principle) have no effect on the impurity.

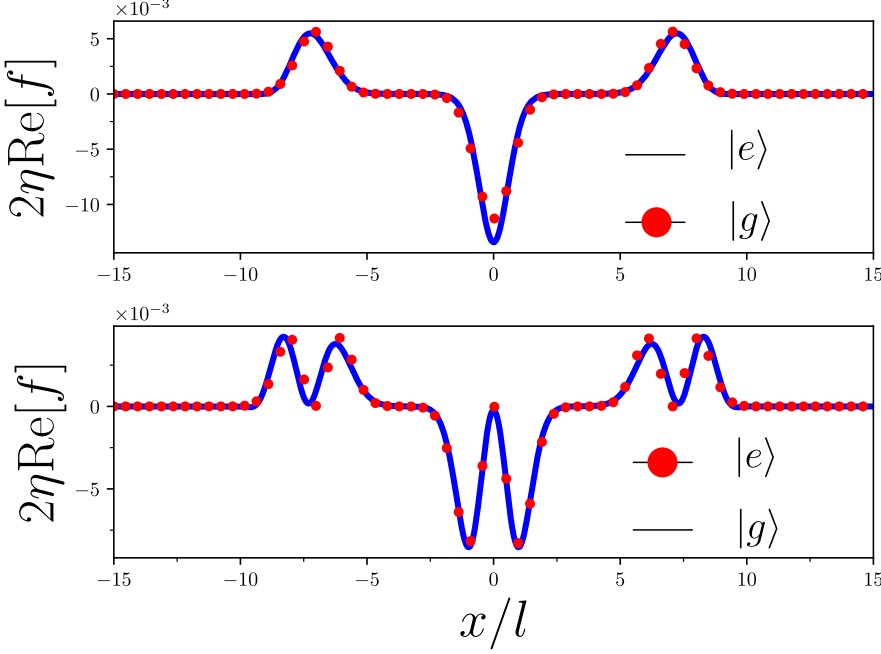

Figure 4: Comparison between the numerical solution of Eqs. (24), (25) (blue line) and the corresponding linear approximation (red dots) for an impurity initialized either in the ground state $|g\rangle$ (top) or the first excited state $|e\rangle$ (bottom) of a harmonic oscillator. Plotted are the densities of the Bose gas at $t \approx 7.1 \left( \frac{ml^2}{\hbar} \right)$. Numerical calculations are implemented with $N = 1500$, $g = 1.0 \left( \frac{\hbar}{ml} \right)$, and $g_{ib} = g$. The density of the Bose gas at $t = 0$ is $\rho = 50/l$.

## 5 Higher Dimensions

We can use the theoretical methods of section 2 to analyze a weakly perturbed Bose gas described by the $d$-dimensional Gross-Pitaevskii equation:

$$i\hbar \frac{\partial \phi_d}{\partial t} = -\frac{\hbar^2}{2m} \frac{\partial}{\partial \vec{x}} \frac{\partial}{\partial \vec{x}} \phi_d + gN|\phi_d|^2 \phi_d + \eta V_d(\vec{x})\phi_d, \tag{28}$$

where $V_d$ is the $d$-dimensional perturbing potential, and $\phi_d$ is the corresponding order parameter. Let us briefly discuss the solution to the GPE in the linear regime. Following Eq. 2, we expand the function $\phi_d$ as in the form

$$\phi_d \simeq \frac{1 + \eta f_d}{L^{\frac{d}{2}}} \exp\left(-\frac{ig\rho t}{\hbar}\right),$$ (29)

where $L$ is the linear dimension of the system, and derive the function $f_d$ as

$$\mathrm{Re}(f_d) = \frac{1}{(2\pi)^d} \int \mathrm{d}^d k \frac{\tilde{V}_d(\vec{k}) e^{-i\vec{k}\vec{x}}}{\epsilon_k + 2\rho g} \left(\cos \omega_k t - 1\right),$$ (30a)

$$\mathrm{Im}(f_d) = -\frac{1}{(2\pi)^d} \int \mathrm{d}^d k \frac{\tilde{V}_d(\vec{k}) e^{-i\vec{k}\vec{x}}}{\epsilon_k + 2\rho g} \left(1 + \frac{2\rho g}{\epsilon_k + \hbar\omega_k}\right) \sin \omega_k t,$$ (30b)

where $k = |\vec{k}|$, $\tilde{V}_d(\vec{k}) = \int V_d(\vec{x}) e^{i\vec{k}\vec{x}} \mathrm{d}^d x$ is the Fourier transform of the potential in $d$ spatial dimensions; by assumption, $\tilde{V}(-\vec{k}) = \tilde{V}(\vec{k})$. The energies $\omega_k$ and $\epsilon_k$ are defined as in section 2.

To illustrate Eqs. (30), let us analyze them in three spatial dimensions in the infrared limit ($k_{pert} \to 0$):

$$\mathrm{Re}(f_3) = -\frac{V_3(\vec{x})}{2\rho g} + \frac{1}{4\pi^2 \rho g} A(\vec{x}, t),$$ (31a)

$$\mathrm{Im}(f_3) = -\frac{1}{2\pi^2 \hbar} \int_0^t A(\vec{x}, T) \mathrm{d}T,$$ (31b)

where

$$A(\vec{x}, t) = \sum_{l=0}^{\infty} \sum_{m=-l}^{l} (-i)^l Y_l^m \left(\frac{\vec{x}}{x}\right) \int_0^{\infty} k^2 \mathrm{d}k \tilde{V}_l^m(k) j_l(kx) \cos(ckt),$$ (32)

here $x = |\vec{x}|$, $Y_l^m$ is the $(l,m)$ spherical harmonic, and $\tilde{V}_l^m(k) = \int \mathrm{d}\Omega Y_l^m \left(\frac{\vec{k}}{k}\right) \tilde{V}_3(\vec{k})$ defines the angular amplitudes of the perturbing potential. The function $A(\vec{x}, t)$, thus also $f_3$, contains only the $(l,m)$ harmonics for which $\tilde{V}_l^m \neq 0$. Therefore, by analyzing the profile of $A$ one can learn about the angular structure of the potential. We are interested in the behavior of $f_3$ far outside the range of the potential, i.e., in the limit $x \to \infty$. Therefore, we approximate the spherical Bessel functions by their asymptotics, $j_l(kx) \simeq \sin(kx - l\pi/2)/(kx)$, which leads to the expression

$$A \simeq \sum_{l,m} \frac{(-i)^l}{2x} Y_l^m \left(\frac{\vec{x}}{x}\right) \int k \mathrm{d}k \tilde{V}_l^m(k) \left(\sin\left(k(x-ct) - \frac{l\pi}{2}\right) + \sin\left(k(x+ct) - \frac{l\pi}{2}\right)\right).$$ (33)

The second term in the integrand is highly oscillating for $x \to \infty$, and we can neglect it. The resulting expression for $A$ then reads as

$$A \simeq \sum_{l,m} \frac{(-i)^l}{2x} Y_l^m \left(\frac{\vec{x}}{x}\right) \int k \mathrm{d}k \tilde{V}_l^m(k) \sin\left(k(x-ct) - \frac{l\pi}{2}\right),$$ (34)

it is clear that $A$ does not vanish only in the region close to $x = ct$, which is what we expect for sound propagation. We leave the study of the general expression for $A$ for future studies. Instead, we focus on a spherically symmetric potential ($l = 0$) for which Eq. (33) is exact for all values of $x$, and can be easily evaluated

$$A(\vec{x}, t) = \pi^2 \left(\frac{x - ct}{x} V_3(x - ct) + \frac{x + ct}{x} V_3(x + ct)\right).$$ (35)

The corresponding three-dimensional density is written as

$$n_3(\vec{x}, t) = \rho_3 + \frac{\eta}{g}\left(\frac{x - ct}{2x}V_3(x - ct) + \frac{x + ct}{2x}V_3(x + ct) - V_3(x)\right), \qquad (36)$$

where $\rho_3 = N/L^3$ is the density of the unperturbed Bose gas. The density $n_3$ has a structure which is very similar to what has been obtained in one spatial dimension: The perturbation creates a defect and a running wave, whose shape is fully determined by the shape of the spherically symmetric perturbing potential. For long times, the term $V_3(x+ct)$ is small, since $V_3$ is an integrable function. In this limit, the density is determined by the defect and the spherical outgoing wave as

$$n_3(\vec{x}, t) \simeq \rho_3 + \frac{\eta}{g}\left(\frac{x - ct}{2x}V_3(x - ct) - V_3(x)\right). \qquad (37)$$

The form of the function $n_3$ implies the possibility to study the perturbing potential by analyzing the generated sound-wave packet, which, however, requires a more complicated procedure than the corresponding one-dimensional analogue.

The physics behind Eq. (36) is similar to that in one spatial dimension. Indeed, the propagation of sound in three spatial dimensions for $l = 0$ can be described by the massless (1+1) Klein-Gordon equation on a semi-infinite line. The corresponding solution is $\frac{v(ct-x)-v(x+ct)}{x}$. The solution should also include a time-independent term that describes the defect in the Bose gas. According to the Thomas-Fermi approximation to the density profile, this defect is given by $-\frac{\eta}{g}V_3(x)$. The form of $v$ then follows from the initial condition, $n_3(t = 0) = \rho_3$.

# 6    Conclusion

To summarize, population of collective modes provides information on the properties of their source. To illustrate this, we have calculated excitations generated in a Bose gas by a static impurity. In the linear regime, these excitations have been expressed through the potential imposed by the impurity. As a relevant example, we have considered a one-dimensional Bose gas with a trapped impurity atom. The impurity has been initialized in either the ground or the first excited states of a harmonic oscillator, although, it is not difficult to argue that our analysis can be extended to more complicated cases. For example, one could measure not only motional but also internal states of an impurity, provided that the impurity-gas interaction strength, $g_{ib}$, depends on the pseudospin of the impurity.

Our work implies that one can hear the shape of a drumstick in a one-dimensional Bose gas in the infrared limit, and motivates to go beyond our analysis. Outside the infrared limit, Eq. (14) poses an invertability problem for the potential $V(x)$ and becomes our next goal. Another future direction is to extend our discussion to higher-dimensional systems with non-spherically symmetric potentials (which might be relevant for molecules in quantum gases [49]), and to assess beyond-mean-field effects which are particularly important for one-dimensional quantum gases.

## Acknowledgements

We acknowledge fruitful discussions with Dr. Simos Mistakidis regarding beyond mean-field effects in our system. We also thank Prof. Maxim Olshanii for valuable suggestions to improve the manuscript.

**Funding information**   O.V.M acknowledges the support from the National Science Foundation through grants No. PHY-1402249, No. PHY-1607221, and No. PHY-1912542 and the Binational (US-Israel) Science Foundation through grant No. 2015616, as well as by the Israel Science Foundation (grant No. 1287/17) and from the German Aeronautics and Space Administration (DLR) through Grant No. 50 WM 1957. This work has also received funding from the DFG Project No.413495248 [VO 2437/1-1] and European Union's Horizon 2020 research and innovation programme under the Marie Skłodowska-Curie Grant Agreement No. 754411 (A. G. V.)

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
