# Peer review of "Shape of a sound wave in a weakly-perturbed Bose gas"

_SciPost Physics_

## Round 1 · Referee Report · Anonymous (Referee 1) · 2020-9-11

Strengths

1) Detailed and precise analysis of the problem, only minor correction at the technical level are required 2) Interesting proposal for an experiment in the ultracold gas context, that is detecting the state of an impurity from the sound waves produced after coupling to a weakly -interacting Bose gas

Weaknesses

1) The main mathematical results themself, for instance Eq. 22, are not very surprising since, within the approximations used, the relevant phonon modes have linear dispersion (i.e. they are dispersionless) and therefore the sound waves are described by a d'Alembert wave equation. These are results that could be easily guessed. On the other hand the authors provide a very detailed and precise derivation of these results in the case of the Gross-Pitaevskii equation.

Report

In their work “Shape of a sound wave in a weakly-perturbed Bose gas”, the authors consider the problem of a one-dimensional weakly-interacting Bose gas which is perturbed at time t = 0 by an external potential and ask the question of what can be learned about the perturbation by analyzing the sound waves produced after the quench. They show that the shape of the perturbing potential defines exactly the shape of the propagating sound wave in the gas (Eq. 22). This result holds in the limit of a weak perturbation, where the nonlinearity of the Gross-Pitaevskii equation modelling the gas can be neglected, and in the limit of long wavelength, that is in the limit where the characteristic scale of the perturbing potential is larger than the healing length of the Bose gas. This latter limit corresponds to neglecting the nonlinearity in the dispersion of the phonon modes in the gas. The results apply also to the case of a 2D or 3D gas if translational invariance is preserved in the directions transverse to motion (in this case beyond mean-field effect not captured by the GPE are also less important). The authors also provide in Section 4 an interesting proposal for an experiment where their ideas could be tested. Finally in Section 5 the authors provide a similar analysis in the 3D case for a spherically symmetric perturbation.

From a mathematical point of view the main result of the work, Eq. 22, is hardly surprising given that within the limits of a weak and long-wavelength perturbation the relevant phonon modes have linear dispersion and the sound waves are effectively described by a d’Alembert wave equation. On the other hand, I think the present work is worth publishing since the authors provide a rather detailed and precise analysis of the problem and also they suggest an interesting experimental protocol that could be tested with current experimental capabilities in the ultracold gas setting. The idea of detecting the initial state of the impurity using sound waves, as discussed in Section 4, is particularly interesting and would be a remarkable and challenging experiment to realize in practice.

Requested changes

Before I can finally recommend the work for publication I suggest that the authors implement some minor changes to their works, as detailed in the points below:

1) clarify this sentence: “Note that the form of the ansatz (2) ensures the non-standard form of the linearized GPE in Eq. (3).” In particular specify in what sense the linearized GPE is in non-standard form. What is the standard form then? 2) explain more precisely why the zero and “lost”mode are negligible as mentioned in this sentence: “There also exist the so-called zero and “lost” modes that are consequences of the Nambu-Goldstone theorem [24, 25] due to the $U (1)$ symmetry breaking [26–30], but their contribution to f(0) is negligible, and we do not consider them here.” 3) Maybe it would be useful to derive explicitly the Klein-Gordon equation from Eq. 3, or at least sketch the derivation in footnote 3. 4) This sentence is probably wrong: “We again consider $\xi << l$, which means that the density distortion created by the impurity has a range, which is smaller than the oscillator length.” I think the condition here is that the harmonic trapping potential is sufficiently shallow so as to ensure that the characteristic size of the perturbation (given by the harmonic oscillator length) is larger than the Bose gas healing length. Therefore one can take the infrared (long wavelength) limit as specified by Eq. 17. 5) Clarify the following sentence: “By changing $g_{ib}$ one creates a sound wave in the Bose gas with amplitude proportional to $g_{ib}$ , whereas the corresponding change of the state of the impurity is of the order $g_{ib}^2$.” In particular spell out the argument showing that the effect of the Bose gas on the impurity occurs only at second order in the coupling parameter.

---

## Round 1 · Referee Report · Anonymous (Referee 2) · 2020-9-15

Strengths

1) The paper is well written and provides sufficient details of the analytical calculations. 2) The problem addressed here is quite simple, but it is analyzed in thorough manner and, as the authors discuss in the Conclusions, it can be extended and generate more refined calculations

Weaknesses

1) The results do not seem groundbreaking, but nonetheless are discussed clearly

Report

In their paper, considering a weakly-interacting one-dimensional quasicondensate, the authors discuss the shape of a wave produced by a perturbation at time t=0.
The perturbation is described by a potential V(x) and the authors derive an analytical formula relating the density at time t>0 (the shape of the wave) with the potential V(x).
The density fluctuation, obtained from the linearized Gross-Pitaevskii equation, is then compared with the numerical solution of the full GPE, finding good agreement.

I like the paper, which offers a simple and thorough study of the propagation of a sound wave. I also appreciate section 4, where a realistic implementation of the problem is considered.

Requested changes

The authors state after Eq. (22) that, since the integral of the real part of f is zero, the density is correctly normalized. I think that the normalization of the wave function should also be discussed immediately after Eq. (2), for instance saying that the integral of the f must be zero.

The paper analyzes a system of particles on a ring, and, due to the boundary conditions, the initial condition is uniform in space.
Could the author discuss what can be expected in the presence of a longitudinal harmonic potential confining the one-dimensional bosonic gas?

For instance, by solving the GPE, it would be interesting to compare the quasicondensate in a ring and the quasicondensate confined in an harmonic potential, whose characteristic length of the harmonic oscillator is taken equal to the circumference of the ring L.

---

## Editorial Decision

resubmitted